# Binocular fusion enhances the efficiency of spot-the-difference gameplay

Kavitha Venkataramanan[1], Swanandi Gawde[1], Amithavikram R. Hathibelagal[1,2], Shrikant R. Bharadwaj[1,2]*

1 Brien Holden Institute of Optometry and Vision Sciences, L V Prasad Eye Institute, Hyderabad, Telangana, India, 2 Prof Brien Holden Eye Research Centre, L V Prasad Eye Institute, Hyderabad, Telangana, India

* bharadwaj@lvpei.org

## Abstract

Spot-the-difference, the popular childhood game and a prototypical change blindness task, involves identification of differences in local features of two otherwise identical scenes using an eye scanning and matching strategy. Through binocular fusion of the companion scenes, the game becomes a visual search task, wherein players can simply scan the cyclopean percept for local features that may distinctly stand-out due to binocular rivalry/lustre. Here, we had a total of 100 visually normal adult (18–28 years of age) volunteers play this game in the traditional non-fusion mode and after cross-fusion of the companion images using a hand-held mirror stereoscope. The results demonstrate that the fusion mode significantly speeds up gameplay and reduces errors, relative to the non-fusion mode, for a range of target sizes, contrasts, and chromaticity tested (all, p<0.001). Amongst the three types of local feature differences available in these images (polarity difference, presence/absence of a local feature difference and shape difference in a local feature difference), features containing polarity difference was identified as first in ~60–70% of instances in both modes of gameplay (p<0.01), with this proportion being larger in the fusion than in the non-fusion mode. The binocular fusion advantage is lost when the lustre cue is purposefully weakened through alterations in target luminance polarity. The spot-the-difference game may thus be cheated using binocular fusion and the differences readily identified through a vivid experience of binocular rivalry/lustre.

## Introduction

In a typical spot-the-difference game (Fig 1), differences in local features of two otherwise identical scenes are detected using an eye scanning (saccadic eye movements) and visual matching strategy. The differences typically used in such images include a polarity change in the local feature, presence/absence of the local feature and shape difference in the local feature (Fig 1). This game is challenging to play for two reasons. First, it imposes a significant cognitive load on the player as they have to scan the scene for a given feature and remember it, saccade to the companion scene and re-scan it for a matching feature and identify changes in the scanned feature, amongst all other distractors. Second, this task is akin to the well-established

**Funding:** This study was supported by the Prof. Brien Holden Eye Research Centre, Hyderabad Eye

Research Foundation. The funders of this study had no role in study design, data collection and analysis, decision to publish, or preparation of the manuscript. The funders of this study supported the salary of AH and SRB. The author(s) have no proprietary or commercial interest in any materials discussed in this article.

**Competing interests:** The authors have declared that no competing interests exist.

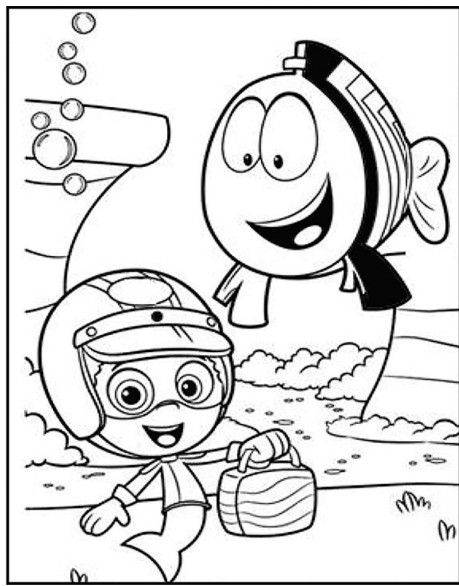

**Fig 1. Sample image pair of the spot-the-difference game.** This image contains 6 differences (position of bubbles, eye-brow of the larger fish, scale of the larger fish, tail fin orientation of the larger fish, image of face in the helmet of smaller fish and rivet in the helmet of smaller fish) used in Experiment 1. This image pair can be cross-fused using convergence eye movements to create the cyclopean percept and experience the fusion mode of gameplay.

change-blindness paradigm in a spatial domain, wherein the visual transient that exists while saccading between the two scenes is thought to render the visual system insensitive and consequently "blind" to the change in the local feature [1, 2].

Apart from the local feature differences, the companion scenes in the spot-the-difference game are otherwise the same and, therefore, very amenable to binocular fusion [3, 4]. Once global fusion is achieved, the local non-fusable features in the resultant cyclopean percept readily exhibit binocular rivalry or luminance/chromatic lustre that will make these features shimmer, amidst other fused features in the scene [3, 4]. Binocular rivalry is the periodic alteration of the fused percept between the two monocular inputs, with periods of mixed-features of both inputs during this alteration process (piecemeal rivalry) [4]. Binocular lustre is a consequence of rivalry that produces a shimmering region of variable brightness [4]. In the spot-the-difference game, polarity differences in the local feature may give rise to binocular lustre while shape differences in the local feature may result in binocular rivalry. Players may simply search for such periodic alterations or shimmering features in the fused image, much like a standard visual search task [5], to identify these differences. In fact, binocular rivalry/lustre has been identified as a strong salience-adding feature that augments the efficiency of visual search [6–8]. Interestingly, the identification of local feature differences has also been demonstrated to become more efficient when the companion images are rapidly presented sequentially in time, relative to when they are placed side-by-side [9]. The binocular fusion equivalent of this task–the present study–has, however, not been explored thus far. Based on these observations and gaps in the literature, we hypothesized that 1) binocular fusion strategy (henceforth referred to as "fusion mode") will increase playing efficiency by completing the task sooner and with lesser errors, relative to the traditional "non-fusion mode" of gameplay and 2) the polarity feature change that elicits the strongest perception of binocular lustre would maximally expedite gameplay in the fusion mode, followed by the shape differences that may result in binocular rivalry, and then the presence/absence feature which may elicit only a weak form of binocular lustre, if any. Addressing these hypotheses constituted the first aim of this study.

The differences in the local features of a scene come in varied sizes, shapes, contrasts and colors in a typical spot-the-difference game (Fig 1). The second aim of the study was therefore to determine if the potential advantage of the fusion mode of gameplay varies with the afore-mentioned target properties of the local feature differences. Feature differences that are larger in size (lower spatial frequencies) [10], higher in contrast [11] and more saturated in chromaticity [12] tend to evoke a more salient sensation of rivalry/lustre and may therefore be hypothesized to have better discriminability in the spot-the-difference game, relative to feature differences in the other end of the spectrum and relative to the non-fusion mode of gameplay.

If, indeed binocular lustre was responsible for the superiority of the fusion mode of gameplay, then this advantage should be lost by weakening the lustre cue. Binocular lustre perception is strongest when the monocular features are of opposite luminance polarity, relative to the background (e.g. a black disc on a gray background in one eye and a white disc on the same gray background in the fellow eye) [13, 14]. Lustre is weakened when the luminance polarity of the monocular features are in the same direction from the background, while maintaining the inter-pair contrast same as the lustre-enhanced pairs (e.g. the two discs in the previous example are both lighter than the gray background, but having different levels of grayscale values that equal in contrast to the previous stimulus). A typical spot-the-difference game has local features that are opposite in polarity and hence expected to produce a strong sensation of binocular lustre (Fig 1). The impact of weakening the lustre cue on the efficiency of spot-the-difference gameplay was also tested in this study by manipulating luminance polarity of the local feature differences, as described above.

Four questions related to the spot-the-difference gameplay were answered in this study. First, is the fusion mode of gameplay more efficient than the non-fusion mode? Amongst the three types of differences available, are polarity differences more prominently identified local feature than other features (Experiment 1)? Second, does the efficiency of gameplay depend on the size and contrast of the local feature differences (Experiment 2)? Similarly, does the efficiency of the two modes of gameplay depend on the chromaticity of the local feature differences (Experiment 3)? Fourth, does the efficiency of fusion mode change when binocular lustre is weakened (Experiment 4)?

## General methods

The study was approved by the institutional review board of Hyderabad Eye Research Foundation, L V Prasad Eye Institute (LEC 01-19-203). The study conformed to the tenets of the Declaration of Helsinki. A total of 100 subjects participated in this study (Experiment 1: n = 39; Experiment 2: n = 21; Experiment 3: n = 20; Experiment 4: n = 20; age range = 18 – 28yrs for all experiments). Different cohort of subjects participated in each of the four experiments, as one single cohort was not available throughout for sequential experimentation. A shortened version of the main experiment (Experiment 1) was however conducted on all participants in Experiments 2–4 to ensure that they showed the same main effect (better performance in fusion compared to non-fusion). All participants were recruited from the student and staff pool of the L V Prasad Eye Institute (LVPEI), Hyderabad, India, after they signed a written informed consent form. All participants were visually normal, free from any ocular pathology, had best-corrected visual acuity of ≤20/20 in both eyes, good binocular fusion capabilities, stereoacuity of ≤40arc sec and correctly identified all color plates in the Ishihara test for color deficiency.

### Experiment I: Is the fusion mode more efficient than the non-fusion mode of gameplay?

**Test stimuli.** A total of 39 subjects (age range: 18 – 28yrs; 18 males and 21 females) participated in this experiment. Twenty-four spot-the-difference image pairs of cartoon characters

were downloaded from the Internet (https://images.app.goo.gl/Zaae8i5eoHhEavDM7) and modified using Adobe Illustrator CC 2015® (Adobe Inc, San Jose, USA) to include the polarity differences in a local feature, presence/absence of a feature and shape difference in a feature (Fig 1). All image pairs were black and white and had ~100% Michelson's contrast [i.e., black features on a white background (Fig 1)]. Each image pair contained 3, 6, 9 or 12 local feature differences that needed to be identified, indicating increasing levels of task complexity. Three different image pairs containing each of the aforementioned number of differences were created, totaling up to 12 image pairs for identification in the fusion and non-fusion modes of gameplay. The three local feature differences were equally distributed in each image (e.g., image pairs containing 6 differences would have two each of the polarity, presence/absence and shape difference). The order of presentation of the image pairs was randomized across trials in each participant to avoid any response bias. The mode of gameplay was randomized only across participants such that there were equal numbers of participants for whom the fusion and the non-fusion modes of gameplay happened first. In total, between the two modes of gameplay, four levels of task complexity and three repetitions of each task complexity, the participants made a total of 180 difference judgments in this experiment [(2 play modes x 3 differences x 3 repeats) + (2 play modes x 6 differences x 3 repeats) + (2 play modes x 9 differences x 3 repeats) + (2 play modes x 12 differences x 3 repeats) = 180 difference judgments].

Image pairs used in the fusion and non-fusion modes of gameplay had to be different to avoid task familiarity. Hence, a pilot experiment was conducted wherein 10 participants identified differences in the 24 image pairs (i.e., 6 image pairs each for 3, 6, 9 and 12 differences) using the traditional non-fusion mode of game play. No significant variation in the task completion time was observed across the image pairs containing a given number of differences in these participants. Therefore, three of these image pairs were then randomly assigned to database for the fusion mode of game play and the remaining three were assigned to the database for the non-fusion mode. These image sets were then used for the first experiment of the study on a completely different set of participants.

**Experimental procedure.** The image pairs were displayed on a luminance calibrated LCD monitor (22"; 1680 x 1050 pixel resolution) using the slideshow option of PowerPoint 2013® (Microsoft Inc., Redmond, USA). All participants were aware of the spot-the-difference game and had played it in the traditional non-fusion mode previously, with 3 observers self-claiming to be "seasoned" at this game based on their frequency of gameplay. Participants were simply instructed to identify the differences in the two companion images by sequentially looking at them in the non-fusion mode of gameplay. In the fusion mode of gameplay, they were instructed to identify regions in the fused percept that appeared "shiny" or "disturbed". Participants identified these differences using mouse clicks in appropriate location of the left-hand size image and were given practice with this task prior to the start of the experiment. Participants were informed of the total number of feature differences present in a given image and also provided immediate feedback for identifying a correct feature difference in the form of a red-cross appearing in the appropriate location of the image. This feedback prevented participants from re-identifying a given feature difference in that image. The time taken to identify a difference was calculated automatically by PowerPoint using its built-in timer functions. The time taken for first difference identification was calculated as the elapsed time between the display of the image and mouse-click in the appropriate location of the image. For subsequent differences, the time taken was calculated as the elapsed time between the previous and the present mouse click for the newly identified feature difference. Mouse clicks made in incorrect locations were ignored. The quickness of mouse clicks could vary across subjects but unlikely to vary across trials within a subject. This factor is therefore unlikely to influence the relative difference in the time taken for task completion across experiments in this study. Image pairs

containing 3, 6, 9 and 12 differences were displayed for a total of 40, 60, 90 and 120sec, respectively. If the subject could not identify a local feature differences within this time, they were considered as misses.

In the non-fusion mode of gameplay, participants performed the task monocularly with their dominant eye (dominance was determined using the Miles test [15]; non-dominant eye was occluded). In the fusion mode of gameplay, participants viewed the image binocularly through a handheld stereo-viewer (Screen-Vu Stereoscope, Portland, USA) that aided fusion of the left and right images into a single cyclopean percept. The cyclopean percept appeared as the central image, flanked by the two monocular percepts. Participants were instructed to ignore the monocular flankers and attend only on the central cyclopean percept. The stereo-viewer is a miniaturized Wheatstone mirror stereoscope that has movable periscopic mirrors to adjust for the participant's latent eye deviations (horizontal phoria) and fuse the monocular percepts [16]. Participants performed the task from 60cm in both modes of gameplay, with their heads stabilized using a chin and forehead rest. A break of ~30min was provided between the two modes of gameplay.

**Data analyses.** The efficiency of gameplay was defined using: 1) the total time taken to identify the local feature differences in a given image (time to complete task) and 2) the number of missed identifications of these differences in a given image. Data analyses were performed using Matlab® R2016a and SPSS Version 20 (SPSS Inc., Chicago, USA). Several pairwise comparisons were made in the data across the four experiments and most of these comparisons met the normality criterion according to the Kolmogorov-Smirnov test. For those comparisons that did not meet this criterion, normality was assumed for ease of statistical interpretation. Therefore, parametric statistics were applied for all comparisons in this study. The time taken for task completion was statistically analyzed in all the experiments using 2- or 3-factor repeated measures ANOVA (RM-ANOVA). Pairwise comparisons were made wherever required using the post-hoc Bonferroni test and appropriate Bonferroni correction for the p-value owing to multiple comparisons have been applied. The effect size was quantified using the partial Eta-squared ($\eta_p^2$) value in SPSS [17]. Homogeneity of variance between data pairs was tested using the Mauchly's sphericity test and a violation of sphericity was corrected using the Greenhouse-Geisser correction factor. The proportion of misses were analyzed using the Chi-squared ($X^2$) test for proportions.

**Results.** The question of whether the fusion mode of gameplay was more efficient than the non-fusion mode was answered affirmatively by comparing the aforementioned outcome variables in the two modes of gameplay across four levels of task complexity and with three different types of local feature differences (Fig 2A and 2B, Experiment 1a in S1 Dataset). A 2-factor RM-ANOVA analysis (gameplay mode x task complexity) indicated a statistically significant large effect size of the gameplay mode [$F_{(1,116)} = 83.8$, $p<0.001$, $\eta_p^2 = 0.42$)] and task complexity [$F_{(3,348)} = 533.7$, $p<0.001$, $\eta_p^2 = 0.822$)] on the time taken for task completion (Fig 2A). The interaction between the two factors was also statistically significant with medium effect size [$F_{(3,348)} = 8.5$, $p<0.001$, $\eta_p^2 = 0.07$)]. Post-hoc Bonferroni test for multiple comparisons indicated that the reduction in time taken for task completion was statistically significant for all four levels of task complexity ($p<0.001$) (Fig 2A and 2B).

The absolute number of misses increased marginally with increasing levels of task complexity (Fig 2C), with these differences being statistically significant only for 9-differences [Chi-squared test of proportions; $X^2_{(1,1053)} = 6.05$, $p = 0.01$] and 12-differences [$X^2_{(1,1404)} = 10.04$, $p = 0.002$] but not for 3-differences [$X^2_{(1,351)} = 1.91$, $p = 0.17$] and 6-differences [$X^2_{(1,702)} = 0.09$, $p = 0.77$]. Since the number of differences that had to be identified also scaled with task complexity, the percentage of misses remained more or lesser similar across task complexity (7.1–13.9% for non-fusion mode and 4.3–10.5% for fusion mode) (Fig 2C).

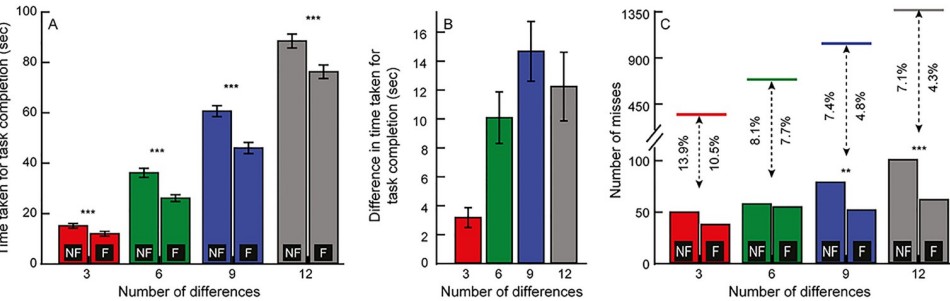

**Fig 2.** Panel A shows the mean (±1 SEM) time taken to complete the task for the four different level of task complexity (3, 6, 9 and 12 differences) in the non-fusion (NF) and fusion (F) modes of gameplay in Experiment 1. Panel B shows the mean (±1 SEM) difference in the time taken between the non-fusion and fusion modes of gameplay for each level of task complexity. Panel C shows the total number of misses of local feature differences across all participants for a given level of task complexity in both modes of gameplay in Experiment 1. The horizontal line above each pair of bars in this panel indicate the total number of differences that had to be identified for that task complexity. The % values for each pair of bars indicate the misses in a percentage scale. Asterisk symbols indicate statistically significant difference between each pair of data obtained in the fusion and non-fusion mode of gameplay. *: p-values between 0.05 and 0.01, **: p-values between 0.01 and 0.001 and ***: p-values <0.001.

The absolute number and percentage of misses were, in general, smaller for the fusion-mode than the non-fusion mode of gameplay (Fig 2C).

A sub-question asked here was about the relative difficulty encountered in identifying the three different types of differences employed in the study. The polarity difference was the most readily identified amongst the three difference types across task complexity in both modes of gameplay, accounting for ~60–70% of all first identified differences in this experiment [fusion mode: $X^2(1,107) \geq 46.9$, p<0.001; non-fusion mode: $X^2(1,101) \geq 24.8$, p<0.001, both relative to the other two difference types] (Fig 3A). The shape and presence/absence differences accounted for ~15–20% of the remaining first identified differences across task complexity in both modes of gameplay [fusion mode: $X^2(1,107) \leq 0.6$, p≥0.44; non-fusion mode: $X^2(1,101) \leq 2.66$, p≥0.10] (Fig 3A, Experiment 1b in S1 Dataset). The proportion of trials in which polarity was identified as the first difference was overall greater in the fusion mode than in the non-fusion mode, across task complexity (Fig 3A, Experiment 1b in S1 Dataset). The time taken to identify polarity, shape and presence/absence as the first difference was shorter in the fusion mode of gameplay than in the non-fusion mode, across task complexity (Fig 3B, Experiment

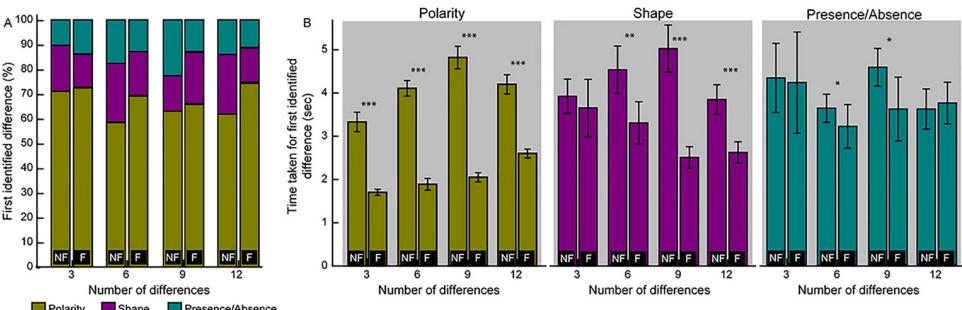

**Fig 3.** Panel A shows the percentage of times a given difference was identified as the first amongst all the possible differences in an image for difference task complexity in Experiment 1 for non-fusion (NF) and fusion (F) modes of gameplay. Panel B shows the mean (±1 SEM) time taken for first correct identification of a difference in a given image for the three types of local feature differences. Given that the polarity difference was overwhelmingly identified as the first in both modes of gameplay, the number of data points that were averaged to compute the mean and SEM in panel B are larger for this difference, relative to the other two. Asterisk symbols are same as Fig 2.

1b in S1 Dataset). This difference was more pronounced for the polarity difference than for the other two difference types (Fig 3B). Three-factor RM-ANOVA (gameplay mode x task complexity x difference type) on the time taken for the first identification showed statistically significant large effect size of gameplay mode [$F(1,9) = 33.6$, $p<0.001$, $\eta_p^2 = 0.79$)] and difference type [$F(2,18) = 49.3$, $p<0.001$, $\eta_p^2 = 0.59$)] but not of task complexity [$F(3,27) = 0.7$, $p = 0.56$, $\eta_p^2 = 0.07$)] (Fig 3B). The interaction between gameplay mode and difference type also approached statistical significance [$F(2,18) = 4.0$, $p = 0.05$, $\eta_p^2 = 0.31$)], confirming the greater impact of binocular fusion in aiding the detection of polarity change, vis-à-vis, the non-fusion mode and the other two difference types (Fig 3B). Other interactions were not statistically significant ($p>0.07$).

Post-hoc Bonferroni test for pairwise comparisons of the main effects revealed the following trends. First, across both modes of game play and the three difference types, the time taken to identify the first difference did not vary statistically with task complexity (mean difference $\pm1SEM \leq \pm0.45\pm0.35sec$; $p\geq0.5$). Second, across both modes of game play and all four task complexities, the time taken to identify polarity difference as the first was significantly different from the time taken to identify shape and presence/absence differences as the first difference (mean difference$\pm1SEM$ at least $-1.2\pm0.26sec$; $p\leq0.006$). The time taken to identify shape and presence/absence differences as the first were not significantly different from each other (mean difference$\pm1SEM = -0.4\pm0.4sec$; $p = 0.9$). Post-hoc testing was not necessary for the main effect of gameplay mode as, to begin with, there were only two comparisons to be made.

Qualitative observation of Fig 3B suggested that the fusion mode of gameplay decreased the time taken for identification for the polarity and shape differences but not for the presence/absence differences, all relative to the non-fusion mode of play (Fig 3B). This pattern was not appropriately captured in the three-factor RM-ANOVA described above for this dataset. Hence, two-tailed, heteroscedastic Student's T-tests were performed with appropriate Bonferroni correction of p-values for multiple comparisons for each pair of fusion versus non-fusion gameplay mode data across different task complexities and difference types. The time taken for first identification was statistically significantly lesser in the fusion mode of gameplay relative to the non-fusion mode, across all task complexities for the polarity difference (mean difference range: 1.60 to 2.77sec; $p<0.0001$) (Fig 3B). The mean difference was smaller but statistically significant for shape differences for images containing 6, 9 and 12 differences (mean difference range: 1.23 to 2.5sec; $p<0.0001$) but not for 3 differences (mean difference: 0.27sec; $p = 0.06$) (Fig 3B). For first identification of presence/absence differences, the difference between the fusion and non-fusion modes of gameplay were different only for images with 6 and 9 differences (mean difference range: 0.42 to 0.97sec; $p\geq0.02$) but not for 3 and 12 differences (mean difference range: 0.11 to 0.14sec; $p\geq0.07$) (Fig 3B).

**Discussion.** The present experiment shows evidence for binocular fusion to improve the efficiency of the spot-the-difference gameplay, vis-à-vis, the traditional non-fusion mode of gameplay. The latter mode of gameplay is a prototypical change blindness task wherein participants' ability to identify changes is close to 50% and, those differences that do get identified, are done so after considerable time is spent scanning the companion images [1, 2]. In this context, the time taken to identify local feature differences in the non-fusion mode of gameplay was quantitatively similar to previous reports.[9, 18]. For instance, the time taken for first difference identification for images with 6 differences was ~5sec in Brunel and Ninio's study [18] and ~3 – 5sec in the non-fusion mode of Experiment 1 in the present study (Fig 3B). The error rate in this study was, however, only 7–13% for image pairs in the non-fusion mode of gameplay (Fig 2B), compared to ~50% error rates reported in previous studies [1, 2]. This difference may be attributed to the more complex real-world stimuli (e.g. museum artefacts [19] or motion pictures [20]) used in typical change-blindness studies where the differences may be more subtle than in the cartoon images used in the present study (Fig 1).

Binocular fusion converts the spot-the-difference game into a visual search task, wherein, the fused percept becomes the visual search scene and the to-be-identified features may stand-out owing to binocular rivalry/lustre, aiding their easy identification [6–8]. Additionally, once fused, local features no longer need to be matched and, therefore, not required to be committed to short-term memory [1, 2], visual transients that render the local feature change ineffective become meaningless [1, 2] and eye movement search strategies get optimized [21], all collectively facilitating change detection. In the present experiment, this was well-reflected as a drop in the time taken by ~50% and a drop in difference identification errors by 35–40%, in the fusion mode of gameplay, relative to the non-fusion mode (Fig 2). Amongst the three local feature differences tested (Fig 1), the polarity difference was most readily identified, relative to the shape or presence/absence differences (Fig 3). This result is quite expected given how the spatial characteristics of the monocular images influence the experience of binocular rivalry/lustre during cyclopean viewing [4]. Monocular images that are opposite in their luminance polarity or very dissimilar in their shape (e.g., orthogonally oriented gratings) tend to generate the strongest experience of lustre and rivalry, respectively [4]. In the images used in this study, the polarity differences were indeed in the opposite direction between the two monocular images (e.g., the fin of the larger fish in Fig 1) but the shape differences were rather subtle in the monocular images (e.g., the pattern of bubbles in Fig 1). In fact, in some of the spot-the-difference image pairs such as the one shown in Fig 1, fusion of the shape differences resulted in a subtle depth perception owing to underlying disparity arising from subtle position differences in the local feature (e.g., cross fusion of Fig 1 will make the bubbles appear in slight depth). These did not necessary stand-out during the cyclopean visual search. In the presence/absence differences, there was really nothing to generate binocular rivalry/lustre and presence of a feature in one image blended in with the absence in the companion image in the fused cyclopean percept (e.g., the eyebrow of the larger fish in Fig 1). Thus, the experience of binocular conflict leading to rivalry or lustre may be strongest for polarity differences, weak for shape differences and very subtle, if any, for the presence/absence differences in the present study. That the non-orthogonal shape differences of the local features in the present experiment (e.g., the pattern of bubbles in Fig 1) are not as strong a stimulus for binocular rivalry as orthogonal stimuli (e.g. orthogonally-oriented Gabor patches typically used to elicit binocular rivalry) is supported by a recent study by Risen et al who showed distinct epochs of fusion and rivalry for the former-type of stimuli than the latter [22]. Perhaps, the shape differences in the present study sometimes elicited an experience of fusion in our participants during the scan sequence, thus making its identification difficult compared to polarity differences.

Performance in the change blindness task could be enhanced by presenting the companion images serially in time, relative to the classic side-by-side presentation [9]. This improvement may be attributed to the reduced oculomotor demand, as there is no need for sequential eye movements between companion images when presented serially in time [9]. In a similar manner, the fusion mode of gameplay too does not require any sequential eye movements between companion images to identify local feature differences and, this reduced oculomotor demand may contribute to the observed enhancement in task performance. Tracking the pattern of eye movements in the fusion and non-fusion modes of gameplay would provide future insights on this issue [5, 21]. It would also be interest in the future to adapt the serial presentation task of Josephs et al into a dichoptic temporal fusion paradigm where the two companion images are rapidly shuffled between the two eyes using a shutter-goggle type apparatus. Perhaps, spotting of local feature differences would be doubly enhanced with the combination of temporal shuffling and binocular fusion of images.

## Experiment 2: Does gameplay efficiency depend on the size and contrast of the local feature difference?

**Test stimuli and experimental procedure.** A total of 21 subjects (age range: 18 – 28yrs; 12 males and 9 females) participated in this experiment. Geometric shapes of 3 different angular subtense (5˚, 3˚ and 1˚ at 60cm viewing distance) and 4 different inter-pair luminance-contrasts (~100%, 50%, 10% and 5%) containing only 6 local feature differences of the polarity type were custom-created using Adobe Illustrator (Fig 4A). The luminance contrast was defined according to Michelson's contrast as the difference in $L_{max}$ and $L_{min}$ grayscale values of the geometric shape between the two-corresponding pair of features (Fig 4A). For instance, a polarity difference with ~100% inter-pair contrast meant that the geometric shape was white (grayscale value of 255) in one image and black (grayscale value of 0) in the corresponding image, against a uniform gray (grayscale value of 127) background (Fig 4A). All contrast values used here were expected to be suprathreshold for these study participants [23]. Feature pairs with no differences were randomly assigned a grayscale value of either of the two images (i.e. either 0 or 255 for the 100% contrast image pair), against a uniform gray background (Fig 4A). With the different target size, contrasts and three repetitions of each, there were a total of 36 image pairs each in the fusion and non-fusion modes of gameplay. In total, participants made 432 difference judgments in this experiment (2 play modes x 3 target sizes x 4 contrast levels x 6 differences x 3 repeats = 432 difference pairs). Since the results of Experiment 1 indicated that shape and presence/absence differences contributed only to a limited extent to the enhancement of gameplay, stimuli with these differences were not included in Experiment 2. Each image pair contained only 6 polarity differences and they were displayed for a maximum of 60sec each, like Experiment 1. The experimental protocol was otherwise identical to that of Experiment 1.

**Results.** Fig 4B–4G show the impact of target size (5˚, 3˚ and 1˚) and contrast (5%, 10%, 50% and 100%) on the time taken for task completion in the two modes of gameplay (Experiment 2 in S1 Dataset). A 3-factor RM-ANOVA (gameplay mode x target size x target contrast) showed the statistically significant large effect size of gameplay mode [$F(1,20) = 6.2$, $p = 0.02$, $\eta_p^2 = 0.24$] on the time taken for task completion. The main effect of target contrast was statistically significant but with a relatively small effect size [$F(3,60) = 3.3$, $p = 0.03$, $\eta_p^2 = 0.14$] while the main effect of target size was not statistically significant [$F(2,40) = 0.18$, $p = 0.84$, $\eta_p^2 = 0.01$]. Amongst the interactions, only the one between gameplay mode and target

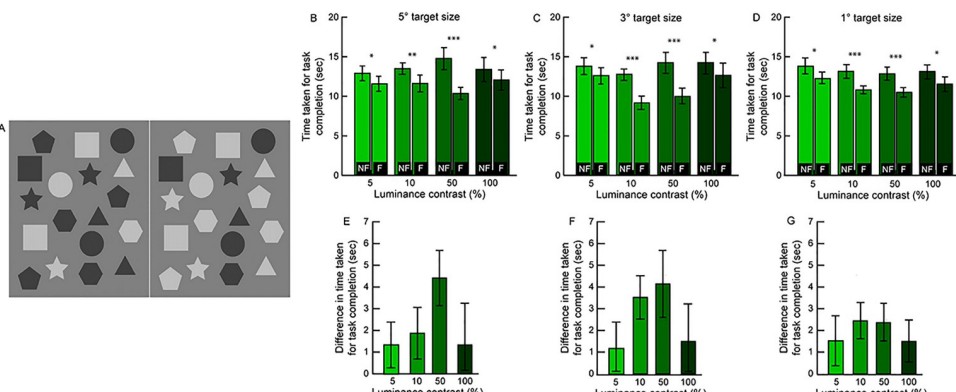

**Fig 4.** Sample grayscale image pair with 50% luminance contrast and 3˚ angular subtense used in Experiment 2 (Panel A). Mean (±1 SEM) time taken for task completion across target contrasts and sizes in the non-fusion (NF) and fusion (F) modes of gameplay in Experiment 2 (panels B–D). Mean (±1 SEM) difference in the time taken between the two modes of gameplay for each combination of target contrast and size (panels E–G). Asterisk symbols are same as Fig 2.

contrast approached significance [F(3,60) = 3.2, p = 0.03, $\eta_p^2$ = 0.14)], while all others remained insignificant (p>0.2). Two-tailed, heteroscedastic Student's T-tests with appropriate Bonferroni correction of p-values for multiple comparisons indicated that the time taken for task completion was statistically significantly lesser in the fusion mode of gameplay relative to the non-fusion mode, across all target sizes and contrasts (p≤0.02) (Fig 4B–4D). Across the three target sizes, the mean difference in time taken between the two gameplay modes was largest for the two intermediate contrast levels (10% and 50%) (p<0.001) than for the lowest (5%) and highest (100%) contrast levels tested here (p<0.02) (Fig 4E–4G).

**Discussion.** Binocular fusion appeared to speed-up the spot-the-difference gameplay, independent of the luminance contrast and sizes of the stimuli tested in this study (Fig 4B–4G). These results suggest that as long as binocular lustre is experienced from the polarity differences in the local features, they will stand-out in the fusion mode of gameplay and become readily amenable for identification, independent of how large the local feature difference is or what the contrast difference is between the local feature pairs. The lack of size dependency in this experiment relates well to the results of Experiment 1 in that the cartoon images used there contained targets of various sizes, some as small as or smaller than the 1° target in the present study (e.g., the polarity difference of the helmet's rivet of the smaller fish in Fig 1). From the present results, it is likely that these small targets in Experiment 1 would have been identified with as much ease as the larger polarity differences targets in the cartoon images (e.g., fin of the larger fish in Fig 1). The effect of target contrast on gameplay performance could possibly be explained by the underlying neurophysiology of how binocular lustre is detected by the visual system [13, 14, 24–26]. The strength of binocular lustre is thought to be mediated by the amount of conflict between neuronal elements in the retina or visual cortex that code for different luminance polarities (e.g., the ON and OFF-centered retinal ganglion cells), when stimulated with luminance patches of opposite polarities [13, 14, 25]. This conflict is expected to increase progressively with an increase in the inter-pair contrast of the luminance patches from 5% to ~100% in this experiment. Accordingly, the salience of lustre experienced and, therefore, the ease of detecting differences in local features based on this cue is expected to be more pronounced with an increase in contrast value. The mean difference in time taken for task completion between the fusion and non-fusion modes of gameplay was smaller for 5% contrast targets, relative to the 10% and 50% contrast targets, for all target sizes (Fig 4E–4G). This is along the lines predicted by the intensity of the lustre experience noted above [13, 14, 25]. Surprisingly, in divergence to the aforementioned prediction, the mean difference for the 100% contrast targets was not as high as those for the intermediate contrast values (Fig 4E–4G). This suggests a ceiling effect for how much binocular lustre can enhance target salience or a reduction in time taken to identify differences in the non-fusion mode of gameplay for this target contrast considering how vivid the inter-pair luminance patches were or an idiosyncratic variation in response that cannot be presently explained. The second factor seems unlikely for the non-fusion gameplay mode data does not reveal any systematic reduction in time taken for task completion for 100% contrast, vis-à-vis, others tested here (Fig 4B–4D). It is also possible that while the strength of the neuronal conflict decreased with a reduction in the inter-pair target contrast, these remained above the threshold required to elicit the experience of lustre and hence did not strongly impact the results at hand. The present experiment did not ask participants to characterize their experience of lustre but simply use this experience to detect local feature differences. Had the former question been asked of the participants, a difference in results between the four-luminance contrasts might have become more remarkable.

As discussed for Experiment 1, performance in a change-blindness type task being dependent on the overall task complexity is also supported by the shorter time taken for task

completion in Experiment 2 (~10 – 15sec), relative to Experiment 1 (~35 – 40sec) (Figs 2A and 4B-4G). In Experiment 1, the stimuli were cartoon images with different types of local feature differences that could appear in any part of the image (Fig 1). On the contrary, the stimuli in Experiment 2 were simple geometric shapes, contained only polarity difference and had only one combination of target size and contrast (Fig 4A). This may have made the task more predictable and easier to complete here than in Experiment 1. A future experiment with multiple combinations of target sizes and contrast in a single image for comparison with cartoon images used in Experiment 1 would address this issue.

## Experiment 3: Does gameplay efficiency depend on chromaticity of the local feature difference?

**Test stimuli and experimental procedure.** A total of 20 subjects (age range: 18 – 28yrs; 8 males and 12 females) participated in this experiment. Geometric shapes of 3˚ angular subtense at 3 different hue directions and at each of four saturation levels (75%, 50%, 25% and 12.5%) containing 6 local feature differences of the polarity type were custom-created using Adobe Illustrator (Fig 5A). Two of the hue axes were along the red-green color directions [140˚ (CIE x, y coordinates at 75% saturation: (0.21, 0.51)– 320˚ (0.36, 0.17)] and 170˚ (0.52, 0.27)– 350˚ (0.18, 0.31)] and the third was along the blue-yellow direction (64˚ (0.39, 0.53)– 244˚ (0.16, 0.06). All target chromaticities were defined with respect to the grey reference point (0.305, 0.323) in the CIE 1931 color space and hue directions were defined counterclockwise with respect to the abscissa [27]. The tristimulus values for each of the hue directions were obtained using an XRite i1 Display Pro Colorimeter (X-Rite, Incorporated, Germany) and they were converted to CIE xyz values [28]. No attempt was made to make the chromatic stimuli equiluminant in this experiment [29]. An image contained feature differences of only one combination of color direction and saturation, all against a uniform gray background (Fig 5A). Feature pairs with no differences were randomly assigned a hue angle and saturation of either of the pair, against a uniform gray background. With the different hue angles, saturations and three repetitions of each, there were a total of 36 image pairs each in the fusion and non-fusion modes of gameplay. In total, participants made 432 difference judgments in this experiment (2 play modes x 3 color directions x 4 saturation levels x 6 differences x 3 repeats = 432 difference pairs). Stimuli were displayed on a luminance plus color-calibrated LCD monitor (24"; 1920 x 1200 pixels resolution) for a maximum of 60sec each. The experimental protocol was identical to that of Experiment 2.

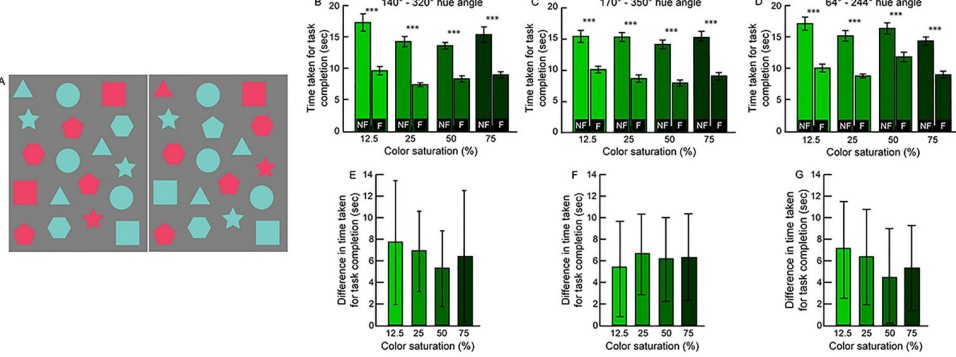

**Fig 5.** Sample image pair along the red-green color axis (140 – 320o hue angle) with 75% color saturation used in Experiment 3 (Panel A). Mean (±1 SEM) time taken for task completion across target color saturations and hue angles (panels B–D). Mean (±1 SEM) difference in the time taken between the non-fusion (NF) and fusion (F) modes of gameplay for each combination of color saturation and hue angle (panels E–G). Asterisk symbols are same as Fig 2.

**Results.** Fig 5B–5G show the impact of color direction (40˚– 320˚, 170˚– 350˚ and 64˚– 244˚) and chromatic saturation (75%, 50%, 25% and 12.5%) on the time taken to task completion in the two modes of gameplay (Experiment 3 in S1 Dataset). A 3-factor ANOVA test (gameplay mode x color direction x chromatic saturation) showed that the time taken for task completion was significantly shorter in the fusion mode than in the non-fusion across all color directions and chromatic saturations [$F_{(1,19)} = 152.9$, $p<0.001$, $\eta_p^2 = 0.89$)] (Fig 5). The main effects of color direction [$F_{(2,38)} = 62.6$, $p = 0.007$, $\eta_p^2 = 0.24$)] and chromatic saturation [$F_{(3,57)} = 5.8$, $p = 0.004$, $\eta_p^2 = 0.23$)] were statistically significant with large effect size. Only the interaction between color direction and chromatic saturation was significant with large effect size [$F_{(6,114)} = 36.4$, $p<0.001$, $\eta_p^2 = 0.20$)] Two-tailed, heteroscedastic Student's T-tests with appropriate Bonferroni correction of p-values for multiple comparisons indicated that the time taken for task completion was statistically significantly lesser in the fusion mode of gameplay relative to the non-fusion mode, across all color directions and chromatic saturations ($p<0.001$) (Fig 5B–5D). Post-hoc pairwise comparisons for color direction revealed that the time taken for task completion in the two red-green color directions (140˚– 320˚ and 170˚– 350˚) were not significantly different from each other (mean difference±1SEM = 0.07±0.30sec; $p>0.5$) but both were significantly different from the blue-yellow color direction (64˚– 244˚; mean difference ±1SEM = 1.01±0.37sec; $p = 0.03$), across all chromatic saturations and modes of gameplay. Post-hoc comparisons for chromatic saturation revealed that the time taken for task completion was significantly different only between the 12.5% and 25% color saturations (mean difference ±1SEM = 1.67±0.45sec; $p = 0.008$), across all color directions and modes of gameplay. None of the other pairwise comparisons were significantly different from each other ($p>0.2$).

**Discussion.** The efficiency of binocular fusion in speeding up the spot-the-difference gameplay was persistent even for chromatic targets of differing chromatic saturations and color directions (Fig 5B–5G). This is not a surprising finding considering that the opposing hues in a given color direction were chosen with equal saturation on either side of the grey reference point in the CIE 1931 color space [27]. Reduction in color saturation was achieved by moving both values in the image-pair closer to this reference point. In this sense, the variations in chromatic saturation used in this experiment were qualitatively similar to the luminance contrasts used in Experiment 2 –a reduction in color saturation and luminance contrast brought the image-pairs close to the grey background in both Experiments 2 and 3, respectively. Interestingly, the impact of binocular fusion on the time taken for task completion appears to be more dramatic for chromatic stimuli than for luminance-contrast based stimuli (Figs 4A and 5A). The mean difference in time taken for task completion between the fusion and non-fusion modes of gameplay ranged between 4.5–7.7sec across color directions and chromatic saturation in Experiment 3 (Fig 5E–5G) while it ranged only between 1.3–4.4sec across luminance contrast and target size in Experiment 2 (Fig 4E–4G). This difference is unlikely to be attributed to the performance differences in different cohort of subjects, because the time taken for task completion in the non-fusion mode was similar in two experiments (~15sec) (Figs 4B–4D and 5B-5D). Larger differences in task performance between the two gameplay modes was therefore due to a shorter time taken in the fusion mode with chromatic stimuli than with luminance stimuli (Figs 4B–4D and 5B-5D). The chromatic stimuli used this experiment were not rendered equiluminant and, thus, the possibility of subjects using lustre generated from the luminance differences between stimuli for identifying the local features cannot be ruled out. However, for the reasons discussed above, it is unlikely that the improvement of task performance in the fusion mode observed here is purely based on luminance lustre, but more so from a combination of luminance and chromatic lustre (Fig 5B–5G). Future studies can isolate these two attributes of the chromatic stimuli to discern their individual contributions to the spot-the-difference gameplay [29].

For any given color direction, the chromatic displacement (i.e., the distance of each colored patch in an image-pair from the reference gray value in the CIE color space [27]) is smaller for stimuli with lower saturations than for stimuli with higher saturations. The chances of experiencing chromatic lustre and, thus, the ease of identifying such image-pairs in the spot-the-difference game has been shown to increase with stimulus saturation [30, 31]. The mechanism governing easier detection of image-pairs with higher chromatic saturations may be speculated to be equivalent to that of increased contrast in the luminance domain, as described in the previous experiment. The time for task completion in the fusion mode of gameplay, however, did not show any such trend in the present study (Fig 5B–5G). Task performance was only about 1- 2sec faster in for stimuli in the red-green color direction than in the blue-yellow direction. One reason for this could be could be attributed to the fact that red/green chromatic direction having lower saturation thresholds than yellow/blue saturation thresholds [32]. Further, like Experiment 2, differences in task performance across chromatic saturations and color directions may become apparent if participants are asked to characterize their experience of chromatic lustre and not simply use it to detect local feature differences.

Lastly, as in Experiment 2 (Fig 4B–4D), the overall time taken for task completion in both modes of gameplay in Experiment 2 was also shorter than what was observed in Experiment 1 (Fig 4B–4D). The explanation for this is also perhaps similar to Experiment 2 in that the same geometric shapes were used in this experiment, albeit colored, and only one combination of color direction and chromatic saturation was present in a given image-pair. This may have made the task cognitively less demanding than what was experienced by subjects in Experiment 1.

## Experiment 4: Is the advantage of fusion mode of gameplay lost with the "weakening" of binocular lustre?

**Test stimuli and experimental procedure.** A total of 20 subjects (age range: 18 – 28yrs; 8 males and 12 females) participated in this experiment. The grayscale image pairs of geometric shapes subtended one of three visual angles (5°, 3° and 1°) at 60cm viewing distance and contained 6 local feature differences (Fig 1D). These were presented each for 60sec in three repetitions on the luminance calibrated monitor as Experiment 1 using the Psychtoolbox interface of Matlab® R2016a (MathWorks, Natick, MA, USA) [33]. Of these differences, 3 feature pairs contained opposite luminance polarity with respect to background (e.g. top left square in Fig 1D, wherein the square in the left panel is lighter than the background while the corresponding square in the right panel is darker than the background) while the remaining 3 feature pairs contained different luminance values but in the same polarity direction with respect to the background (e.g. top center circle in Fig 6A, wherein both circles are lighter than the background, but the lightness of the one in the right is more than that of the left). Feature pairs with opposite polarity are meant to enhance the experience of lustre while those with same polarity are meant to weaken the experience of lustre [13]. The inter-pair contrast was constant (~15%) in both types of feature pairs, ensuring that differences in task efficiency between stimuli were dependent only on the luminance polarity and not on the inter-pair contrast difference [13]. Feature pairs with no difference were randomly assigned a grayscale value of either of the two images, against a uniform gray background (Fig 6A). With the different target sizes and three repetitions of each, there were a total of 9 images each in the fusion and non-fusion modes of gameplay. In total, participants made 108 difference judgments in this experiment (2 play modes x 3 target sizes x 6 differences x 3 repeats = 108 difference pairs). The experimental procedure was identical to that of Experiment 1. However, unlike Experiments 1–3, only the time to task completion was considered as the outcome variable in Experiment 4. The time

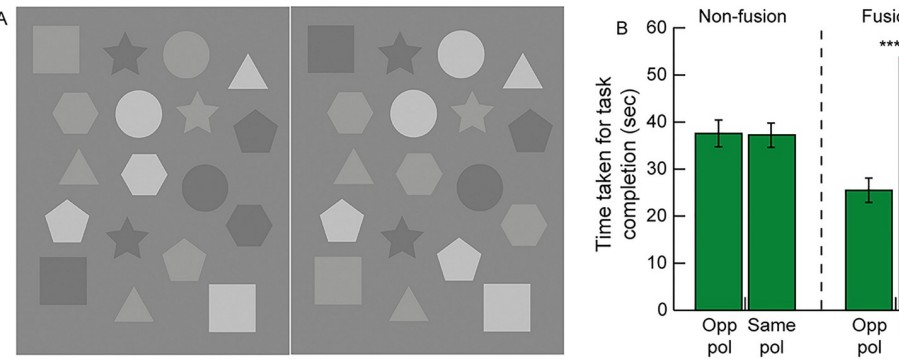

**Fig 6.** Sample image pair used in Experiment 4 (Panel A). This image contains difference pairs with opposite luminance polarity that is meant to enhance the experience of lustre (top left square, middle-right hexagon and bottom left square) and difference pairs with same luminance polarity that weakens the experience of lustre (top circle, middle hexagon and bottom-right hexagon). Mean (±1 SEM) time to task completion for difference pairs with opposite luminance polarity (Opp pol) that enhances the experience of binocular lustre and difference pairs with same luminance polarity (Same pol) that weaken the experience of binocular lustre (Panel B). Each pair of bars show data from the non-fusion and fusion modes of gameplay. Asterisk symbols are same as Fig 2.

taken to identify a difference was calculated automatically by Matlab using its built-in timer functions.

**Results.** The time to task completion for local feature differences that had opposite luminance polarity that enhanced lustre was compared with the corresponding time for local feature differences with the same luminance polarity that weakened lustre (Fig 6B, Experiment 4 in S1 Dataset). Three different target sizes were tested in this experiment but the data were similar for all of them [e.g., mean (±1SEM) time to task completion in the fusion mode of gameplay for difference pairs containing opposite luminance polarity was 24.2±2.0sec, 26.8 ±3.4sec and 25.5±2.2sec for 1˚, 3˚ and 5˚ target sizes, respectively]. Hence, they were combined for statistical analysis (Fig 6B). The time taken for task completion was lesser for targets with opposite polarity in the fusion mode of game play, relative to targets with the same polarity in the fusion mode and relative to both the same and opposite polarity targets in the non-fusion mode of gameplay (Fig 6B). On the contrary, the time taken to complete task was significantly higher for targets with same polarity in the fusion mode of game play, relative to both the same and opposite polarity targets in the non-fusion mode of gameplay (Fig 6B). The time taken to task completion was not different between the same and opposite polarity targets in the non-fusion of game play (Fig 6).

These results were confirmed by a two-factor RM-ANOVA test that showed no significant main effect of mode of gameplay [$F(1,19) = 1.5$, $p = 0.22$, $\eta_p^2 = 0.003$)] but a significant main effect of inter-pair polarity difference [$F(1,19) = 65.4$, $p<0.001$, $\eta_p^2 = 0.11$)] and a significant interaction between the two factors [$F(1,19) = 72.1$, $p<0.001$, $\eta_p^2 = 0.12$] on time task completion. The significant interaction between the two main factors reflected the differential impact of inter-pair polarity difference on the time taken to complete task in the fusion and non-fusion modes of game play shown in Fig 6B. Two-tailed, heteroscedastic Student's T-tests with appropriate Bonferroni correction of p-values for multiple comparisons indicated that the data for same and opposite polarity targets were significantly different from each other only in the fusion mode of gameplay ($p<0.001$) but not in the non-fusion mode of gameplay ($p = 0.86$).

**Discussion.** By weakening the luster cue through the luminance polarity manipulation described above [13, 14], the experiment demonstrated a two-fold increase in the time taken to complete the difference identification, relative to the lustre enhanced condition, both in the fusion mode of gameplay (Fig 6B). This result verified the hypothesis that the experience of

rivalry/lustre is key to speeding-up gameplay in the fusion mode. In the absence of lustre, such as in local feature differences with same polarity, the difference "blends into" the fused portions of the image as a stable percept and do not get identified at all. This blending-in may be the result of a weak or no conflict between neuronal elements in the retina or visual cortex that code for opposite luminance polarities, when the inter-pair polarity of the local feature difference was in the same direction, relative to when they were of opposite polarity [13, 14, 25]. In theory, if this blending-in was complete, the participants of this experiment should not have identified more than 3 local feature differences (the ones corresponding to the opposite polarity). However, other than a few participants who missed the identification, majority of them were able to identify all the six differences, albeit, taking much longer in this task (Fig 6B). One possible explanation for this result is that the blending-in of the lustre weakened difference pairs were not complete but severely weakened, as observed previously for similar values of inter-pair luminance polarity [26, 34]. The task is thus made significantly harder but not impossible through this manipulation. In fact, this can be demonstrated by cross-fusing the companion images in Fig 6A, wherein the local feature differences with opposite luminance polarity readily stand-out while those with the same luminance polarity only exhibit a subtle shimmering effect (e.g., circle in the top right and pentagon in the bottom right). All participants were made aware that there was a maximum of 6 differences in each image pair and, this knowledge may have encouraged to search for differences, however subtle. Had this information been withheld from the participants, a greater proportion of them may have missed the local feature differences with similar polarity.

The time taken for task completion in the fusion mode with opposite polarity inter-pair differences was ~12sec faster than the non-fusion mode of gameplay (Fig 6B). This difference is very much in line with the results in Experiments 1–3 (Figs 2A and 2B, 4B-4D and 5B-5D). Manipulating the lustre cue did not make any difference to the time taken for task completion in the non-fusion mode of gameplay (Fig 6B). This was expected because, unlike the fusion mode that appears to rely on the experience of lustre, participants simply look for differences in a given attribute of the local feature (luminance, contrast, chromaticity, shape, orientation, etc) between the companion images in the non-fusion mode of gameplay. In Experiment 4, the difference in inter-pair luminance, irrespective of their polarity relative to the background, was obvious and well above the detection thresholds for these participants (Fig 6A). Participants would have therefore adopted the same strategy to identify the differences in the non-fusion mode, irrespective of whether the inter-pair difference was of the same polarity or opposite polarity.

## General discussion and conclusions

Improved detection of interocular changes in position (disparity), luminance or chromaticity is a fundamental advantage of processing visual information binocularly [35], and this study has demonstrated an application of this maxim in augmenting "difference" identification in the fusion mode of spot-the-difference gameplay (Figs 2–6). This advantage appears to be largely dependent on the experience of luminance or chromatic lustre and independent of the size, contrast, color direction and chromatic saturation of the local feature differences in the range tested here (Figs 4–6). Four caveats to the fusion mode of this gameplay are worth pointing out here. First, some versions of the spot-the-difference game arrange the companion images up and down, as opposed to horizontally. Such a vertical arrangement is not amenable to binocular fusion owing to limited vertical vergence capability [36, 37]. The advantage of binocular fusion may therefore be restricted to only horizontal arrangements of the companion images during gameplay, unless of course, the player chooses to rotate the image by 90˚ for fusion. Second, not all color combinations when fused produce chromatic rivalry [38, 39]. The

hue angles and saturations were carefully chosen in this study and the experiment was also run in a color-calibrated monitor to ensure a strong perception of chromatic lustre (Fig 5) [38]. Thus, improved efficiency of gameplay owing to chromatic lustre might not be guaranteed for colored image pairs created casually for the purposes of game-play. Besides, the luminance cues may be still present which could also contribute to some of the variability in the results noted in the third experiment. Third, the advantage of fusion to gameplay rests on the fundamental assumption that participants can superimpose the companion images into a single and clear percept. The hand-held stereoscope used for fusion here is unlikely to be available with players during casual gameplay. They will therefore have to rely on voluntarily crossing their eyes to fuse the image pairs and keep their focusing relaxed to maintain image clarity, much like what one would do while viewing an autostereogram [40]. Unfortunately, this is not an innate ability and requires significant training [40–42]. Fourth, for reasons similar to point 3, one-eyed individuals and those with impaired binocular vision (e.g. amblyopia [43, 44]) may not experience the advantage of fusion mode of gameplay. Such a difficulty has been demonstrated in amblyopia for a real-world scene matching paradigm that resembles a change blindness task [45].

An augmentation of change detection through binocular fusion has applications beyond just the gameplay explored here. Despite significant advances in computer technology, human intervention continues to play a paramount role in identifying subtle differences between companion images in the areas security surveillance [46] or disease progression using scan reports [47]. These difference identifications may be made more efficient if the images that are being compared are fused into a cyclopean percept. Further, recent vision therapy for developmental neuro-ophthalmic disorders like amblyopia rely on dichoptically-presented computer games that encourage the amblyopic eye's participation through binocular fusion [48, 49]. The spot-the-difference game could be adapted into a dichoptic presentation and used as engaging stimuli for children undergoing such therapy.

## Supporting information

**S1 Dataset.**
(XLSX)

## Author Contributions

**Conceptualization:** Kavitha Venkataramanan, Swanandi Gawde, Amithavikram R. Hathibelagal, Shrikant R. Bharadwaj.

**Data curation:** Kavitha Venkataramanan, Swanandi Gawde, Amithavikram R. Hathibelagal, Shrikant R. Bharadwaj.

**Formal analysis:** Kavitha Venkataramanan, Swanandi Gawde, Amithavikram R. Hathibelagal, Shrikant R. Bharadwaj.

**Funding acquisition:** Shrikant R. Bharadwaj.

**Investigation:** Kavitha Venkataramanan, Amithavikram R. Hathibelagal, Shrikant R. Bharadwaj.

**Methodology:** Swanandi Gawde, Amithavikram R. Hathibelagal, Shrikant R. Bharadwaj.

**Project administration:** Shrikant R. Bharadwaj.

**Resources:** Shrikant R. Bharadwaj.

**Software:** Kavitha Venkataramanan, Shrikant R. Bharadwaj.

**Supervision:** Shrikant R. Bharadwaj.

**Validation:** Shrikant R. Bharadwaj.

**Visualization:** Swanandi Gawde, Amithavikram R. Hathibelagal, Shrikant R. Bharadwaj.

**Writing – original draft:** Kavitha Venkataramanan, Swanandi Gawde, Amithavikram R. Hathibelagal, Shrikant R. Bharadwaj.

**Writing – review & editing:** Kavitha Venkataramanan, Swanandi Gawde, Amithavikram R. Hathibelagal, Shrikant R. Bharadwaj.

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
