## [Decision Letter · Decision Letter 0]

6 Apr 2021

PONE-D-21-02432

Binocular fusion enhances the efficiency of spot-the-difference gameplay

PLOS ONE

Dear Dr. Bharadwaj,

Thank you for submitting your manuscript to PLOS ONE. After careful consideration, we feel that it has merit but does not fully meet PLOS ONE’s publication criteria as it currently stands. Therefore, we invite you to submit a revised version of the manuscript that addresses the points raised during the review process.

Two expert reviewers have assessed your work. Both reviewers find the study and it's results interesting, but both note that the manuscript is at times lacking in scientific rigor. I particularly agree with both authors that the binocular mechanisms which underlie your findings need to be better discussed. Reviewer 1 further raises some methodological issues which should be addressed, and suggests that portions of the manuscript should be re-written to better fit a scientific publication. Finally, I cannot find anywhere the data underlying your results. Please note that Plos ONE policy requires your raw data to be publicly available, so you should either provide these data as supplementary information or upload them to a public data repository (e.g. Zenodo) and include the link to the deposit in your manuscript. 

We look forward to receiving your revised manuscript.

Kind regards,

Guido Maiello

Academic Editor

PLOS ONE

Journal Requirements:

2) We note that you have included the phrase “data not shown” in your manuscript. Unfortunately, this does not meet our data sharing requirements. PLOS does not permit references to inaccessible data. We require that authors provide all relevant data within the paper, Supporting Information files, or in an acceptable, public repository. Please add a citation to support this phrase or upload the data that corresponds with these findings to a stable repository (such as Figshare or Dryad) and provide and URLs, DOIs, or accession numbers that may be used to access these data. Or, if the data are not a core part of the research being presented in your study, we ask that you remove the phrase that refers to these data.

3)  Thank you for stating the following financial disclosure:

 [The funders had no role in study design, data collection and analysis, decision to

publish, or preparation of the manuscript.].

Reviewers' comments:

Reviewer's Responses to Questions

**Comments to the Author**

1. Is the manuscript technically sound, and do the data support the conclusions?

Reviewer #1: Yes

Reviewer #2: Yes

2. Has the statistical analysis been performed appropriately and rigorously? 

Reviewer #1: Yes

Reviewer #2: No

3. Have the authors made all data underlying the findings in their manuscript fully available?

Reviewer #1: Yes

Reviewer #2: No

4. Is the manuscript presented in an intelligible fashion and written in standard English?

Reviewer #1: Yes

Reviewer #2: Yes

5. Review Comments to the Author

Reviewer #1: This is a nice paper that makes a cute point about how spot-the-difference games can be quickly solved via binocular fusion of the difference-image pairs. Although the study does not provide any new theoretical insights into the mechanisms for detecting interocular differences in contrast, it has the merit of potentially appealing to a broad audience given the popularity of spot-the-difference gameplay. The paper is well-written, the methods sound and the results convincing. My only suggestion for improvement is that it would have been nice if the authors could say a little more about what are believed to be the underlying mechanisms for the detection of interocular contrast differences and what they think are the ecological situations in which these mechanisms are deployed by vision. There have been a number of recent papers on the subject of interocular contrast difference detection which a quick search will reveal and these could form the basis of a more general discussion of the topic.

Reviewer #2: Stats are ok but incomplete. I didn't check for data availability.

Other comments are in the attached file

just typing to reach the minimum character limit in the system................................

6. PLOS authors have the option to publish the peer review history of their article (what does this mean?). If published, this will include your full peer review and any attached files.

Reviewer #1: No

Reviewer #2: **Yes: **Alexandre Reynaud

---

## [Author Response · Author response to Decision Letter 0]

20 May 2021

Please see attached "Response to reviewers" document

---

## [Decision Letter · Decision Letter 1]

21 Jun 2021

PONE-D-21-02432R1

Binocular fusion enhances the efficiency of spot-the-difference gameplay

PLOS ONE

Dear Dr. Bharadwaj,

Thank you for submitting your manuscript to PLOS ONE. After careful consideration, we feel that it has merit but does not fully meet PLOS ONE’s publication criteria as it currently stands. Therefore, we invite you to submit a revised version of the manuscript that addresses the points raised during the review process.

Reviewer 2 has confirmed that you have successfully addressed all their previous comments. Given the substantial restructuring of your paper, I think your final Discussion and Conclusion section is now appropriate for the journal. The reviewer has also provided some final suggestions regarding the interpretation of your study that I would ask you to consider and incorporate in your manuscript. I don't think these minor points should be controversial to address, so I will most likely assess your final revisions directly, with no need for further review. I look forward to receiving your revised manuscript. 

We look forward to receiving your revised manuscript.

Kind regards,

Guido Maiello

Academic Editor

PLOS ONE

Journal Requirements:

Reviewers' comments:

Reviewer's Responses to Questions

**Comments to the Author**

1. If the authors have adequately addressed your comments raised in a previous round of review and you feel that this manuscript is now acceptable for publication, you may indicate that here to bypass the “Comments to the Author” section, enter your conflict of interest statement in the “Confidential to Editor” section, and submit your "Accept" recommendation.

Reviewer #2: All comments have been addressed

2. Is the manuscript technically sound, and do the data support the conclusions?

Reviewer #2: Yes

3. Has the statistical analysis been performed appropriately and rigorously? 

Reviewer #2: Yes

4. Have the authors made all data underlying the findings in their manuscript fully available?

Reviewer #2: (No Response)

5. Is the manuscript presented in an intelligible fashion and written in standard English?

Reviewer #2: Yes

6. Review Comments to the Author

Reviewer #2: I appreciate the efforts made by the authors and the new rearrangement of the manuscript. I still think the general conclusion reads more like a guide on how to optimally play a game than a scientific article. But I will let the editors judge if those are in line with the editorial policies of the journal. Otherwise, I only have minor issues regarding some points which I think should be discussed a bit more thoroughly in the sub-discussions parts.

I think some hypothesis why no effect of contrast is observed for both luminance and color conditions should be discussed.

Why the detection in the color condition is faster than the luminance condition should be discussed as well.

Specific points:

Line 52: I don’t think people would play a game if they think it is laborious.

Line 431: “The strength of binocular lustre is thought to be mediated by the amount of neuronal conflict between the ON and OFF-centered ganglion cells in the retina, when stimulated with luminance patches of opposite polarities.” And Line 636 “This blending-in may be the result of a weak or no neuronal conflict between the ON- and OFF-centered retinal ganglion cells when the inter-pair polarity of the local feature difference was in the same direction, relative to when they were of opposite polarity.” This is more likely to originate in V1 than in the retina.

Line 547: “The chances of experiencing chromatic lustre and, thus, the ease of identifying such image-pairs in the spot-the-difference game has been shown to increase with stimulus saturation and is expected to be somewhat lesser in the red-green color direction than in the blue-yellow direction.” Why?

Line 671: you can’t say it is independent on those parameters. It is only independent in the range you tested.

7. PLOS authors have the option to publish the peer review history of their article (what does this mean?). If published, this will include your full peer review and any attached files.

Reviewer #2: No

---

## [Author Response · Author response to Decision Letter 1]

29 Jun 2021

Please see attached "Response to reviewers" document.

---

## [Editor Report · Decision Letter 2]

2 Jul 2021

Binocular fusion enhances the efficiency of spot-the-difference gameplay

PONE-D-21-02432R2

Dear Dr. Bharadwaj,

We’re pleased to inform you that your manuscript has been judged scientifically suitable for publication and will be formally accepted for publication once it meets all outstanding technical requirements.

Kind regards,

Guido Maiello

Academic Editor

PLOS ONE
---

## [Editor Report · Acceptance letter]

12 Jul 2021

PONE-D-21-02432R2 

Binocular fusion enhances the efficiency of spot-the-difference gameplay 

Dear Dr. Bharadwaj:

I'm pleased to inform you that your manuscript has been deemed suitable for publication in PLOS ONE. Congratulations! Your manuscript is now with our production department. 

Kind regards, 

on behalf of

Dr. Guido Maiello 

Academic Editor

PLOS ONE